# Evidence-based teaching practices correlate with increased exam performance in biology

**Sungmin Moon**, **Mallory A. Jackson**, **Jennifer H. Doherty**, **Mary Pat Wenderoth** *

Department of Biology, University of Washington, Seattle, Washington, United States of America

☯ These authors contributed equally to this work.

* mpw@uw.edu

**Data Availability Statement:** All relevant data are within the paper and its Supporting information files.

## Abstract

Evidence-based teaching practices are associated with improved student academic performance. However, these practices encompass a wide range of activities and determining which type, intensity or duration of activity is effective at improving student exam performance has been elusive. To address this shortcoming, we used a previously validated classroom observation tool, Practical Observation Rubric to Assess Active Learning (POR-TAAL) to measure the presence, intensity, and duration of evidence-based teaching practices in a retrospective study of upper and lower division biology courses. We determined the cognitive challenge of exams by categorizing all exam questions obtained from the courses using Bloom's Taxonomy of Cognitive Domains. We used structural equation modeling to correlate the PORTAAL practices with exam performance while controlling for cognitive challenge of exams, students' GPA at start of the term, and students' demographic factors. Small group activities, randomly calling on students or groups to answer questions, explaining alternative answers, and total time students were thinking, working with others or answering questions had positive correlations with exam performance. On exams at higher Bloom's levels, students explaining the reasoning underlying their answers, students working alone, and receiving positive feedback from the instructor also correlated with increased exam performance. Our study is the first to demonstrate a correlation between the intensity or duration of evidence-based PORTAAL practices and student exam performance while controlling for Bloom's level of exams, as well as looking more specifically at which practices correlate with performance on exams at low and high Bloom's levels. This level of detail will provide valuable insights for faculty as they prioritize changes to their teaching. As we found that multiple PORTAAL practices had a positive association with exam performance, it may be encouraging for instructors to realize that there are many ways to benefit students' learning by incorporating these evidence-based teaching practices.

## Introduction

Implementing active learning in college science, technology, engineering, and math (STEM) classrooms significantly improves student academic performance as compared to passive

**Funding:** Support for this study was provided to authors M.P.W. and J.H.D. by the National Science Foundation (https://www.nsf.gov/) DUE 1725149 and TUES 1118890. Any opinions, findings, and conclusions or recommendations expressed in this material are those of the author(s) and do not necessarily reflect the views of the National Science Foundation. The funders had no role in study design, data collection and analysis, decision to publish, or preparation of the manuscript.

**Competing interests:** The authors have declared that no competing interests exist.

lecture [1–5]. Furthermore, active learning diminishes academic achievement gaps in exams and pass rates for minoritized STEM undergraduates, and this effect is maximized if active learning is implemented at a high intensity [6]. The President's Council of Advisors on Science and Technology found that the US economy will require one million more STEM professionals than the current education system is on track to produce [7]. This same report indicated that over the past decade, 750,000 additional STEM degrees could be earned if retention rates for STEM students were merely increased from the current 40% to 50%. By enhancing learning and decreasing gaps in academic performance, not only will retention rates increase, but a greater diversity of students will have an opportunity to earn a STEM degree [8].

Based on these papers and this report, there have been calls from the National Science Foundation, the American Association for the Advancement of Science, and others for faculty to increase their efforts to incorporate these effective teaching methods in their courses [9–15]. However, active learning encompasses a wide range of teaching methods and research has shown that all are not equally effective at enhancing student performance [6, 16, 17]. If we are to encourage faculty to transform their STEM classes by implementing more active learning methods, it is necessary to identify specific teaching methods that are correlated with academic performance and determine the predicted change in exam performance with increased implementation of the teaching methods.

The term active learning represents multiple teaching techniques from the simple "muddiest point", "peer instruction", and "clicker questions" to more complex techniques of problem-based learning and team-based learning [18–23]. Given that these named teaching techniques encompass multiple teaching practices and a wide range of formats of student and faculty interactions, an alternative strategy for discerning the basis of effective teaching is to identify features of the techniques that may be the basis for their effectiveness.

We have previously developed a classroom observation tool, Practical Observation Rubric To Assess Active Learning (PORTAAL), that assesses the presence, intensity, and duration of different elements of effective classroom teaching practices [24]. Multiple factors impact student learning, including but not limited to classroom teaching practices, factors associated with the student's motivation and background, the way in which the instructor scaffolds the learning of the topic, frequency of pre-class preparation assignments, availability of practice exams and homework assignments, and alignment of course material with formative and summative assessments. Though all these factors may impact student learning, previous studies have shown that classroom teaching practices are a major contributor [5, 6, 25].

The teaching practices included in PORTAAL were based on evidence from the primary literature that when only that practice was added to a current teaching method, student academic performance improved [24]. As the active learning teaching practices identified in PORTAAL have data to support their effectiveness, they are by definition evidence-based teaching practices. However, the practices included in PORTAAL may not include all evidence-based teaching practices, as this is an evolving field of study. Therefore, in this paper we will refer to the specific evidence-based teaching methods included in PORTAAL as PORTAAL practices.

Each of the practices identified in PORTAAL is based in a theoretical framework of learning [24]. Most are based on constructivism and social cognitive theory, both of which are summarized in the Interactive, Constructive, Active, Passive (ICAP) theory of learning [26]. The ICAP framework is empirically grounded and supported by evidence [26, 27]. The ICAP theory stipulates that there is a significant difference between "learning activities" instructors select for their students to do in the classroom and "engagement activities" which refer to the way a student engages with the learning activities. In the framework, a Passive mode of engagement involves activities where students are receiving information. A typical example of this would be students sitting and listening to a lecture or video without any other overt activity. If

the student takes notes during the class or manipulates objects as part of a task, then the engagement activity is categorized as Active. Their definition of active differs from the commonly used term "active learning" which has a more expansive and amorphous interpretation that can include note taking, but often refers to activities that have the learner more deeply engaged with solving problems or synthesizing content [28].

The final two engagement modes are Constructive and Interactive. Constructive activities require the learner to generate an output and include but are not limited to the following activities: asking questions, posing problems, generating predictions, and reflecting and monitoring one's understanding. Interactive activities are defined as activities where two or more students iteratively and constructively discuss a topic. Chi and Wylie hypothesized and found support for their claim that Constructive activities maximize student learning as they produce deep understanding with potential for transfer while the Interactive mode produces deeper understanding with potential to innovate novel ideas [26]. As we compared the PORTAAL practices to the teaching practices categorized in the ICAP framework, we found a strong alignment of the majority of PORTAAL practices with the Interactive and Constructive levels of the ICAP. Therefore, we have used the ICAP learning theory to frame our thinking as to why some practices are helpful for improving student exam performance.

To determine the efficacy of classroom teaching practices on student learning, a means to assess that learning must be identified. Learning is a process that occurs in the student's mind and is invisible to others but can be assessed by determining what students can do with their learning. One such measure of what students can do with their learning are course exams. Ideally, course exams are generated by the instructor to assess student mastery of stated course objectives, but there is evidence that this goal is not always met [29, 30]. However, as grades determine if students will advance to the next level of their academic curriculum and ultimately graduate from a science major, we will use exam scores as a proxy for student academic performance.

Exam questions vary in level of cognitive challenge [31] and numerous studies have found that the academic challenge level of course exams often falls at the lower level of cognitive challenge [29, 32, 33]. We will use Bloom's Taxonomy of Cognitive Domains [31] to categorize questions and determine the cognitive challenge of instructor's exams [34, 35]. The Bloom's level of exams can then be used as a proxy for the cognitive challenge level of exams, as students would be expected to perform better on low Bloom's versus high Bloom's exam questions [36]. As the learning objectives on course syllabi of most faculty indicate that students will be expected to work at higher cognitive levels [29, 37], it will be important to not only determine which PORTAAL practices correlate with student exam performance overall, but which practices correlate with student performance on cognitively challenging exams. To gain greater insight as to how the intensity and duration of each PORTAAL practice correlates with student performance at different levels of cognitive challenge, we will also analyze student performance on exams at lower and higher Bloom's levels.

In addition to improving the cognitive skills of students, it is important to ensure that teaching methods are equitable for all students. Previous research has shown that, despite the numerical dominance of women in biology, women underperform on exams as compared to men with similar college grade point averages. Specifically, research findings indicate that men outperform women on high Bloom's level exams even after controlling for prior academic ability [38, 39]. Students from underrepresented minority groups in STEM or students who are economically or educationally disadvantaged also underperform on exams [6, 40]. If we can identify PORTAAL practices that have a differential correlation with student academic performance by demographic group, it could help decrease this discrepancy.

We will conduct a retrospective study to investigate the correlation between the implementation of PORTAAL practices used in biology classrooms and student performance on exams. We will use the PORTAAL classroom observation tool to conduct a fine-grained analysis of the intensity or duration of PORTAAL practices that correlate with improved student exam performance, first while controlling for Bloom's level of exam and student demographic variables, and then looking specifically at how these practices are associated with performance on exams at low and high Bloom's levels. We will also quantify the expected change in exam score with the addition of each instance of a significant practice. This level of detail will provide valuable insights for faculty as they prioritize changes to their teaching.

To this end, we pose two research questions:

1. Which PORTAAL practices correlate with student exam performance in biology courses at a Research 1 university and do the practices show any bias by demographic group?

2. Which PORTAAL practices correlate with student performance on exams at low and high Bloom's levels?

## Materials and methods

### Participants

**Instructors and courses.** This research is a retrospective study of the teaching methods used by faculty at a major Research 1 university in the Northwest. This study examined 33 biology faculty and 10 different undergraduate biology courses at over a period of four academic years. At this university, large-enrollment classes are recorded using lecture capture technology installed in the classrooms and these recordings are archived. Faculty were not involved any formal professional development and were not specifically trained in the use of PORTAAL practices. Faculty had varied levels of experience using active learning, but information about experience was not collected.

There were four lower-division courses, each of which were offered multiple times across the four years, and six upper-division courses, each offered only once. Three of the lower division courses were either team-taught or taught by a single instructor depending on the offering; the other lower-division course and all upper-division courses were taught by a single instructor. When the course was taught by two instructors, each instructor taught half of the ten-week quarter and gave half of the exams. As the instructors, exams, Bloom's level of exams, and students' exam performance were different in each section of a team-taught course, each section was treated as independent and became a "unit" of analysis. The solo-taught courses were each their own unit. In total, we had 40 units of lower-division courses and 6 units of upper-division courses to analyze. These 46 units were taught by 33 faculty, as some instructors taught two units. Instructor and course information is described in S1 Table and Table 1.

**Table 1. Instructor and course information.**

|  | Lower Division | | Upper Division | Total |
|---|---|---|---|---|
| *n* Courses | 4 | | 6 | 10 |
|  | Team-taught | Solo-taught | Solo-taught |  |
| *n* Offerings | 16 | 10 | 6 | 32 |
| *n* Units of Analysis | 30 | 10 | 6 | 46 |
| *n* Students | 11,218 | 3,415 | 323 | 14,956 |

**Students.** There were 14,956 student data points across all courses (Table 1). Introductory courses ranged from 200 to 700 students, whereas upper-division courses ranged from 24 to 120 students. The registrar provided students' demographic information, which included self-identified binary gender, grade point average (GPA) at the start of the term, participation in the educational opportunity program (EOP) for students identified by the university as economically or educationally disadvantaged, and whether the students self-identified as belonging to a race/ethnicity that is traditionally underrepresented in science (underrepresented minority or URM: e.g., Hispanic, Black/African-American, Pacific Islander, Native Hawaiian, Native American).

Across students in our sample, 58.9% of students (n = 8,808) self-identified as female, 9.6% of students (n = 1,436) self-identified as a race/ethnicity underrepresented in science (URM), and 16.6% of students (n = 2,490) were classified as eligible for the university's educational opportunity program (EOP). For our analysis, we will use eligibility for the EOP as a proxy for socioeconomic status. This research was done under the approval of the University of Washington IRB protocols 38945 and STUDY00002830. Verbal consent was obtained from participants when required by the IRB protocol.

## Data collection

**Classroom observation data.** We randomly selected three to four archived videos from each of the 46 units for a total of 150 videos (S2 and S3 Tables). To select which days to include in this study, the 10-week quarter was divided into four equal time periods excluding the first and last week of the quarter, and one day was randomly selected from each time period. For team taught courses, three videos (except in one instance when four were coded) were randomly selected from weeks two through five for the first instructor and six through nine for the second instructor. Based on a Stains and colleagues [41] study that suggested that at least four observations are necessary for reliable characterization, we coded four classroom videos from all units of analysis collected after the publication of that study.

All videos were coded by trained researchers using the classroom observation tool, POR-TAAL [24]. The PORTAAL coding rubric is found in (S1 Fig). Pairs of researchers independently coded each video of the entire class session before they met to reconcile differences and come to complete consensus. Due to the fine-grained nature of the PORTAAL rubric and that every second of a class session is given at least one code, there is no single PORTAAL score for each class session. Therefore, we were not able to calculate inter-rater reliability between coders. The most common coding disagreements were Bloom's level of the activity, time differences of a few seconds when recording the start and end of an iteration or activity, organizing the activity into a new iteration of the same activity or a new activity, or tallying the number students who responded during debrief if student voices were difficult to hear. Coders discussed these disagreements and came to complete consensus on PORTAAL values for each video. Prior to the statistical analysis, we determined an average value for each PORTAAL practice for a unit of analysis by totaling the number or duration of each PORTAAL practice across all videos for each unit of analysis and dividing by the number of videos. The average values for each PORTAAL practice were then standardized to a 50-minute period which is the length of most class sessions at this institution.

**Exam data.** Most courses had three or four exams per quarter. In team-taught courses, each instructor usually gave two exams. Exams were in multiple formats, including but not limited to the following: multiple-choice, fill-in-the-blank, true/false, and constructed response questions. We collected exam questions with keys that indicated the point value of each question. We also collected the total exam score for each exam for each student. As each unit was

taught by a different instructor and each instructor had exams worth a varying number of points, students' exam scores for each exam were normalized to 100 points. The normalized exam scores for each student were totaled and averaged to create an overall exam score for each student. As this is a retrospective study, instructors could not change the composition of exams in these courses to alter student academic performance.

## Bloom's categorization of exam questions

To attempt to control for the varying cognitive challenge of exams across all courses, we categorized each exam question according to Bloom's Taxonomy of Cognitive Domains [31, 34]. We realize that all attempts at categorizing exam questions have limitations. The challenge level of the exam can be influenced by many factors including but not limited to the following: the length and format of the question, wording of the question stem and distractors [42], cognitive load of the question [43], and the alignment of the test with the level of instruction delivered [44–46].

Based on previous research findings [34, 36, 37], we find Bloom's Taxonomy a reasonable proxy for cognitive challenge level which we realize can differ from performance level. We acknowledge that questions at the lower level of Bloom's taxonomy can be quite difficult for students and lead to lower performance [29] as these questions can address minutia or require knowledge of very specific and often obscure differences between answer choices. However, previous research indicates a negative correlation between Bloom's level and student exam performance [36]. Given these findings, it is important to determine how teaching methods correlate with student exam performance on exams at both low and high Bloom's levels. As most faculty indicate on their syllabi that they want students working at higher cognitive levels in their courses [29], it is important to determine which teaching practices provide the best type of practice needed for students to excel at these higher levels of cognition.

Two trained researchers (authors M.P.W. and M.A.J.) with expertise in biology independently coded all exam questions using the six levels of Bloom's Taxonomy from a Bloom's level 1 (knowledge) to Bloom's level 6 (evaluation). The researchers then met to reconcile any differences in their codes and come to one-hundred percent consensus. Following methodologies used in our previous papers [36, 38], the Bloom's levels were collapsed from six to three: low (knowledge and comprehension), medium (application and analysis), and high (synthesis and evaluation). The three-level Bloom's code and the point values for each question were then used to generate a weighted Bloom's score for each exam question [36, 38]. We pooled the weighted Bloom's scores across questions for all exams in each unit of analysis to produce a total weighted Bloom's score. This score was normalized using the total possible weighted Bloom's score for that unit of analysis [36]. We converted the weighted Bloom's scores to a 100-point scale as in Freeman et al. [36] and Wright et al. [38]. Given our sample size, we could not parse and analyze three Bloom's levels of exams. Therefore, we grouped units into either low or high Bloom's level exams in the structural equation models, using the median value of the units (median = 53.8) as the cutoff. A histogram of the weighted Bloom's scores for all units is illustrated in S2 Fig.

## Latent profile analysis

We used latent profile analysis (LPA) in Mplus 8.3 [47] to determine if the patterns of intensities or durations of the PORTAAL practices were consistent across the observations of each unit of analysis. If the latent profiles were consistent across all observations for each unit, we calculated a mean value of each PORTAAL practice. These mean values were then used to examine correlations between each PORTAAL practice and student exam scores with

structural equation modeling (SEM). If the latent profiles for a unit of analysis were not consistent across all observations, data from that unit were removed from further data analysis.

In our first analysis, we conducted the LPA with 150 daily observations from 46 units of analysis to categorize the level of implementation of 21 PORTAAL practices. We selected the number of latent profiles of PORTAAL practices following a stepwise process that combined statistical model-fit indices with model usefulness indicators (S4 Table) following Wang and Wang [48]. This process was based on classification quality and theoretical underpinnings related to the substantive interpretability of the profiles. The best-fit model produced two profiles representing instructors who implemented PORTAAL practices at either a high or low level. Of the 21 PORTAAL practices, seven practices (Table 2) did not produce profile probability values in the LPA (i.e., the model did not converge), possibly due to a lack of variation. We conducted the LPA again on all 150 daily observations but with only the 14 PORTAAL practices that produced probability values. Of the 46 units of analysis, 42 units were categorized as being at the same profile across all observations (i.e., "consistently high" or "consistently low" profiles), whereas four units showed "mixed" profiles across the three or four

**Table 2. PORTAAL practices.**

| Dimension | PORTAAL Practices | Duration (D)/Instance (I) |
|---|---|---|
| **Practice** | (1) Total time (minutes) students were thinking, working, talking [TST] | D |
| | (2) *n* times the instructor prompted use of prior knowledge [PK] | I |
| **Logic Development** | (3) *n* times high Bloom's activities in class [HB] | I |
| | (4) *n* times students thought alone before answering [Alone] | I |
| | (5) *n* times students worked in small groups [SG] | I |
| | (6) Amount of time in debrief [DB] | D |
| | (7) Amount of time that students talked in debrief [ST_DB] | D |
| | (8) *n* times a student volunteer answered [Vol_Ans] | I |
| | (9) *n* times instructor explained answer [Ins_Exp] | I |
| | (10) *n* times a student volunteer explained the answer [Vol_Exp] | D |
| | (11) *n* times a student explained their answer [Exp_Ans] | I |
| | (12) *n* times an alternative answer was explained [Alt_Ans] | I |
| **Accountability** | (13) *n* times a student random call answered [RC_Ans] | I |
| **Reducing Apprehension** | (14) *n* times the instructor gave a student positive feedback [PFBS] | I |
| **Practices not included in SEM** | (15) *n* multiple-choice questions [MCQ] | I |
| | (16) *n* short answer questions [SA] | I |
| | (17) *n* times instructor answered [Ins_Ans] | I |
| | (18) *n* times whole class answered [WC_Ans] | I |
| | (19) *n* times a student random call explained the answer [RC_Exp] | I |
| | (20) *n* times faculty prompted students to explain their logic [Prom_Log] | I |
| | (21) *n* times the instructor gave the class positive feedback [PFBC] | I |

Practices from PORTAAL that improve student academic performance based on evidence from the literature. Practices are clustered in four dimensions. Seven practices were removed before conducting SEM. Practices were coded as either intensity (instances) or duration (minutes).

observations. Therefore, we included only the 42 units with consistent profiles across all observations and only the 14 PORTAAL practices used to create those two profiles in the structural equation models.

## Structural equation modeling

We used structural equation modeling (SEM) path analysis to investigate if any of the 14 PORTAAL practices correlate with student academic performance. SEM implies a structure for the covariances between the observed variables and allows for investigation of causality and coordination of multiple factors impacting an independent variable. Path analysis is an approach to modeling explanatory relationships between observed variables. Within the path analysis framework, independent variables are assumed to have no measurement error, whereas dependent variables may contain residual terms. Residual terms are the parts left unexplained by the independent variables. For example, other factors we have not measured and/or that are outside our variables of interest can impact academic performance (e.g., scheduling issues with other required courses in their major, students' work hours, students' preference for a certain course or instructor, or instructors' exam grading methods).

Student exam scores, the dependent variable, were regressed on PORTAAL practices, the independent variables. To minimize the confounding effects of cognitive challenge of exams and students' academic preparation, we included weighted Bloom's score of exams and students' GPA at the start of the term as covariates in the SEM. We also included all student demographic variables (gender, EOP status, and URM status) and the interactions between the student demographic variables as covariates in the model. Since previous research has consistently shown that achievement gaps exist by demographic groups in undergraduate STEM courses [49], we were also interested in using SEM with mediation to investigate the effects of PORTAAL practices mediated by demographic factors (gender, EOP, and URM). We did not find any significant effects mediated by EOP or URM, as there was a limited sample size of EOP and URM students which resulted in an imbalanced comparison between EOP and non-EOP and URM and non-URM students. We were able to detect effects of PORTAAL practices mediated by binary gender which could be a result of a more balanced sample of female and male students in our dataset. The full model that was tested can be seen in S3 Fig.

We calculated a correlation matrix of the 14 PORTAAL practices to determine if there was a multicollinearity problem with including multiple independent variables. We found correlation values higher than 0.8 in six pairs of practices, but these practices did not emerge as significant predictors of student exam performance in the SEM (S5 Table). Therefore, there was no multicollinearity problem [50].

To answer research question two, we conducted separate SEMs for units with exams at low and high Bloom's levels. We followed the same procedure as above, regressing student exam scores on PORTAAL practices, using students' GPA, students' demographic variables, and the interactions between those variables as covariates. We also tested for the mediated effects of gender, EOP status, and URM status.

## Results

### Research question 1: Correlation between PORTAAL practices and exam performance

Four PORTAAL practices, small group activities, random call, explaining alternative answers, and total student time, were significantly associated with higher student exam performance while controlling for Bloom's level of exam, students' GPA at the start of the term, EOP status,

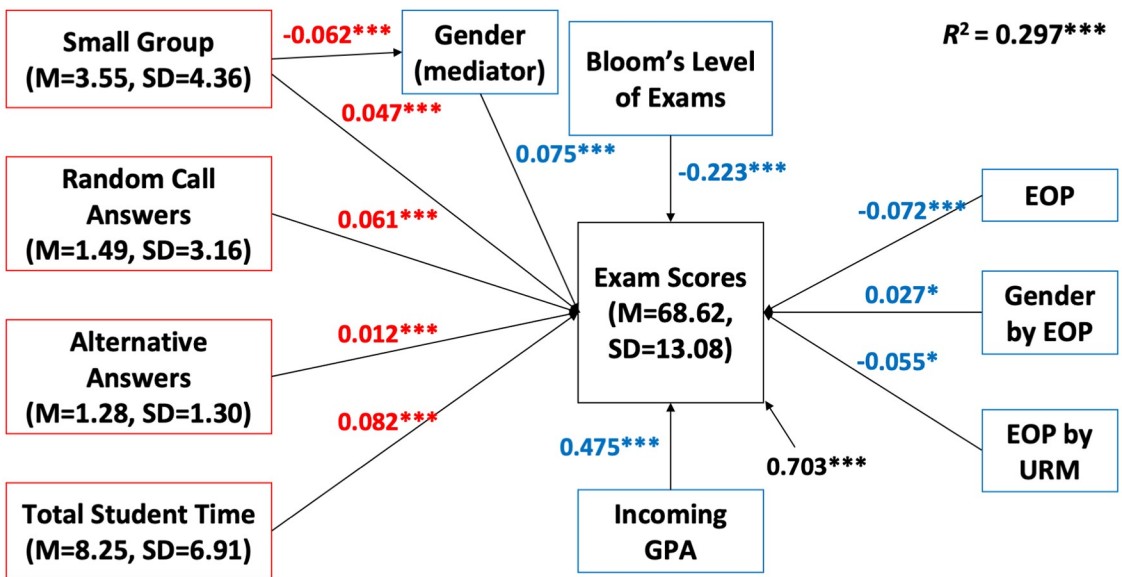

**Fig 1. Structural equation model path diagram of PORTAAL practices for all units of analysis.** Standardized path coefficients are in red. The effects of covariates and interactions are in blue. Residual variances in exam scores not explained by this model are in black. When controlling for the effects (in blue) of covariates and interactions, student exam scores would change by 0.047 standard deviations (0.047×13.08 = 0.61) given a one standard deviation change in the number of small group activities (4.36) while all other evidence-based teaching practices were held constant. $n$ = 42. Significant relationships are marked with $^*$ in this diagram. $^*p < 0.05$, $^{**}p < 0.01$, $^{***}p < 0.001$.

URM status, and interactions between the student demographic factors (Fig 1). Additionally, small group activities had a mediated effect on student exam performance by gender such that small group activities offered an additional benefit to women. The predicted percentage point increase on exams that students would be expected to earn if their instructor implemented on average one more of the designated PORTAAL practice per 50-minute class session is shown in Table 3.

To calculate the predicted percentage point increases for each significant PORTAAL practice, we divided the standard deviation of the exam scores by the standard deviation of the PORTAAL practice and multiplied it by the path coefficient for that practice. Using small group activities as an example, exam scores had a standard deviation (SD) of 13.08 percentage points and the SD for number of small group activities was 4.36. Therefore, the change in exam score per small group activity is 13.08/4.36 = 3 standard deviations. The path coefficient from the SEM analysis indicates that exam scores will change by 0.047 SD per small group activity, therefore, the actual change in exam score is 0.047×3 for an increase of 0.14 percentage points for each small group activity per 50-minute class (all values are found in Fig 1). Though the percentage point increase is small, instructors used an average of 3.55 small group activities per class and the effect is multiplied for each instance of small group work.

These results indicate that students would be expected to earn 0.14 percentage points more on exams if one more small group activity per 50-minute class session was used, 0.25 percentage points more if one more random call was used per class session, 1.18 percentage points more if one more alternative answer was explained per class session, and 0.16 percentage points more on exams if one more minute students thought, worked, or talked per class session (Table 3). In a class with 500 exam points, this would result in 0.7, 1.25, 5.9, and 0.8 additional points if one additional type of each practice was used during each class session. There

**Table 3. The relationships between the change in PORTAAL practices and the expected change in exam scores for analyses of both research questions.**

| PORTAAL Practice | Research Question 1 | Research Question 2 | |
|---|---|---|---|
| | All Exams | Low Bloom's Exams | High Bloom's Exams |
| Small Group | 0.14*** | 0.39*** | - |
| | 0.1***(F) | 0.21* (F) | - |
| Random Call Answers | 0.25*** | -0.71*** | - |
| Alternative Answers | 1.18*** | 1.32*** | 0.36** |
| Working Alone | - | -0.54*** | 0.43* |
| Total Student Time | 0.16*** | 0.35*** | - |
| Student Time in Debrief | - | 0.73* (M) | - |
| Explaining Answers | - | - | 0.20*** |
| | - | - | 0.8** (F) |
| Positive Feedback | - | - | 0.24** |

Expected change in exam scores (percentage points) predicted by one-unit (instance or minute) change in each practice. Mediators are in parentheses (F: female, M: male). Bloom's level of exam, GPA, EOP status, URM status, and interactions between the student demographic factors are included as covariates. Cells with a dash indicate no significant correlation.

$^*p < 0.05$,

$^{**}p < 0.01$,

$^{***}p < 0.001$.

was also an effect that was mediated by gender. If the number of small group activities increased by one per 50-minute class session, female students' exam scores would be expected to increase by 0.15 percentage points.

The SEM explained 29.7% of the variance in student exam scores. All the significant relationships are shown in Table 3 and Fig 1. Since the coefficients are standardized, we can say the predictive power of explaining alternative answers is stronger than the other three practices.

## Research question 2: Correlation between PORTAAL practices and exam performance on low and high Bloom's exams

We used separate structural equation models to determine which PORTAAL practices correlated with exam scores in courses with low Bloom's level exams and courses with high Bloom's level exams.

In units of analysis with lower Bloom's level exams on average, three PORTAAL practices were significantly positively correlated with student exam scores: small group activities, explaining alternative answers, and total student time. Two PORTAAL practices showed a significant negative correlation with exam scores: random call and students working alone. Student time in debrief was significantly associated with an increase in exam scores for male students, while small group activities were associated with an increase in exam scores for female students. The SEM for units of analysis with lower Bloom's level exams explained 28.3% of the variance in student exam scores. All the significant relationships are shown in Table 3 and Fig 2. Since the coefficients are standardized, we can say the predictive power of total student time is stronger than that of the other four practices in the low Bloom's level SEM.

In units of analysis with exams on average at higher Bloom's levels, four PORTAAL practices were associated with increased exam scores: explaining alternative answers, students working alone, students explaining answers, and instructors giving positive feedback to

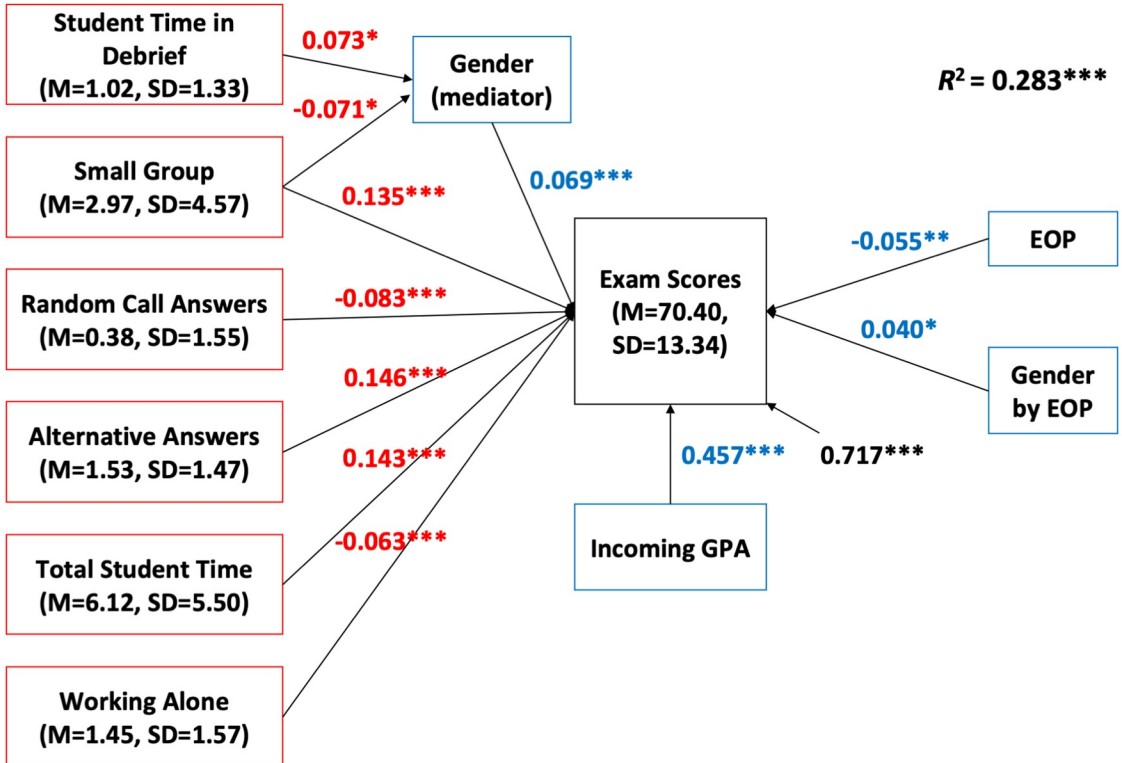

**Fig 2. SEM diagram for units with low Bloom's level exams.** Standardized path coefficients are in red. The effects of covariates and interactions are in blue. Residual variances in exam scores not explained by this model are in black. $n = 22$. Significant relationships are marked with * in this diagram. *$p < 0.05$, **$p < 0.01$, ***$p < 0.001$.

students. Students explaining answers offered an additional benefit to female students. The SEM for units with higher Bloom's level exams explained 31.1% of the variance in student exam scores. All the significant relationships are shown in Table 3 and Fig 3. Since the coefficients are standardized, we can say the predictive power of students explaining answers is stronger than the other three practices in the high Bloom's level SEM.

## Discussion

Our study is the first to demonstrate a correlation between the intensity or duration of implementation of PORTAAL practices (i.e., evidence-based teaching practices) and student exam performance. Though previous research has shown the positive impact of incorporating active learning in the STEM classroom on student exam performance [5, 6], none have provided a fine-grained analysis to identify the correlation between specific classroom teaching practices and exam performance. Of the PORTAAL practices included in the SEM, four were associated with predicted increases in exam scores across all units of analysis while controlling for Bloom's level of exam and students' GPA at start of the term: implementation of small group activities, using random call to solicit answers to in-class questions, explaining alternative answers to in-class questions, and total time that students are actively engaged in class (thinking alone, working in small groups, offering answers). In courses with exams at lower Bloom's levels, three of these practices were associated with increases in exam scores, while exam performance was predicted to decrease when random call and students working alone were used. In courses with exams at higher Bloom's levels, explaining alternative answers correlated with

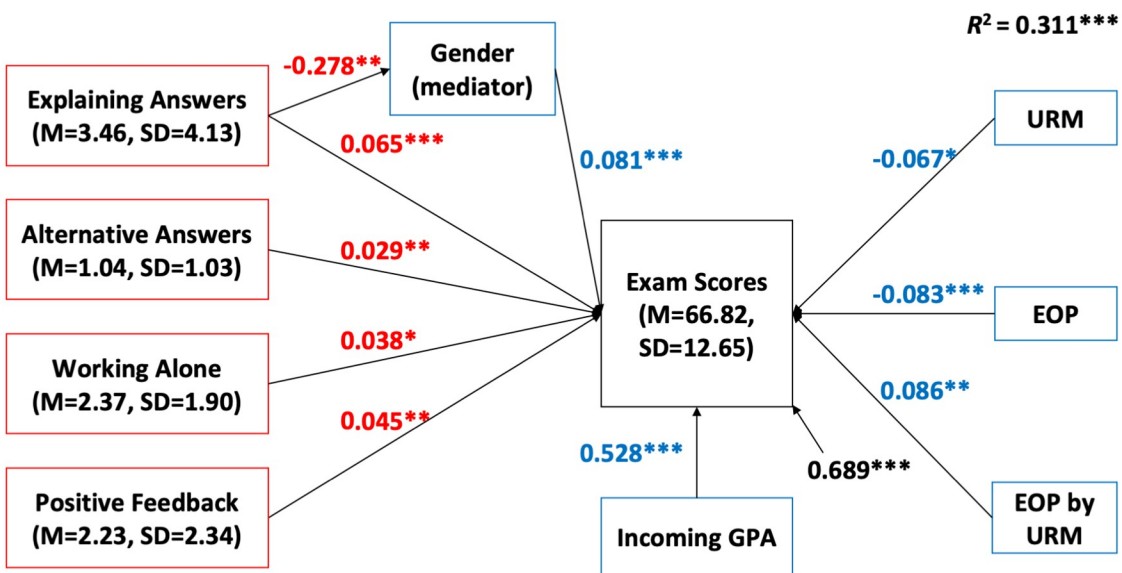

**Fig 3. SEM diagram for units with high Bloom's level exams.** Standardized path coefficients are in red. The effects of covariates and interactions are in blue. Residual variances in exam scores not explained by this model are in black. $n = 20$. Significant relationships are marked with * in this diagram. $^*p < 0.05$, $^{**}p < 0.01$, $^{***}p < 0.001$.

increased exam score, but in contrast to classes with low Bloom's exams, the model predicted that working alone, students explaining answers, and instructors giving positive feedback to students would increase exam scores.

Though the predicted percent increases in exam scores per instance of each PORTAAL practice appear small, the practices are often implemented multiple times and in combination during a single class period. Furthermore, each additional exam point could be the difference between letter grades, for example between a C and a C-. Harris and colleagues [8] found that underrepresented students were more likely to persist in STEM majors if they received a grade of a C or higher and conversely left STEM if they received a C- or lower in their first term Chemistry course. Therefore, these small changes may be the difference between retaining or losing a more diverse STEM population.

### PORTAAL practices associated with improved exam scores

**Small group activities.** We found that use of small group work was positively correlated with student exam performance across all units of analysis when controlling for Bloom's level of exams. We coded a 'small group' activity when students were asked to work with one or more peers to discuss a prompt, whether it was a clicker question, a formal worksheet, or an impromptu question generated by the instructor. The ICAP framework suggests that class activities that provide students with the opportunity to be Interactive and Constructive are effective at improving learning [26], and our results provide empirical support of that claim. Chi and Wylie [26] categorize small group activities as Interactive as this practice provides students with the opportunity to discuss and defend their answers with peers. Similarly, Andrews and colleagues [51] hypothesized that effective learning activities require students to generate their own understanding which is the goal of most small group work. Our finding is aligned with many previous studies that demonstrate the improvement in learning that students gain from working in small groups [52–56]. Small group work may be an added benefit in classes that have more academically challenging exams as the small group work may give students the

additional practice with and peer feedback on explaining and developing the logic and reasoning skills needed for increased performance on these types of exams.

While the use of small group activities was positively correlated with all students' academic performance, the model predicted that this PORTAAL practice offered a small additional benefit to female students in courses with exams at lower Bloom's levels. Small group work supports greater social cohesion between members of a large lecture class [57]. This cohesion can provide a safe learning environment which may contribute to female students' willingness to engage in learning activities [58]. This additional benefit to women may contribute, even in a small way, to eliminating the documented gender gap in student academic performance [38, 59]. We were surprised not to detect a similar mediated effect at higher Bloom's levels, but this may be due to a limited amount of variation in our sample or the effects of other PORTAAL practices on exams at higher Bloom's levels.

**Random call.**   Across all units of analysis, randomly calling on students to answer a question was positively correlated with student exam performance when controlling for Bloom's level of exams. We coded 'random call answers' when a student or group of students gave an answer to a question when called on by the instructor using a randomly generated list of names, seat numbers, or group names. Using random calls to solicit answers from students can be a form of accountability [24]. In classrooms where instructors regularly call on volunteers, some students may put less effort into answering questions if they are accustomed to other students volunteering an answer. When students know that their name or their groups' name may be called on to answer, they put more effort into solving the posed question [60]. Research has also found that regular use of random call can increase all students' frequency of voluntary participation and comfort with participating in class discussions [61]. As student volunteers also tend to be male [59, 62], this leads to a bias in student voices heard in the classroom. Using random call can therefore enhance equity and inclusion as all students have an equal chance of being called on.

However, using random call was significantly negatively correlated with student performance when exams were at lower Bloom's levels. Research has shown that random call can induce some anxiety in students, which may decrease students' sense of safety in the classroom and negatively impact exam performance [63, 64]. There are multiple ways to implement random call that may alleviate the anxiety [64] and instructors need to determine the procedure that best matches the characteristics of their student population. As instructors in these courses had a low incidence of random call (mean = 0.38 times per class, SD = 1.55), it is possible that students did not have enough opportunities to become comfortable sharing their answers with the whole class, and instead felt anxious about random call. However, this result could also be due to not enough variation in our sample of courses with low Bloom's exams to detect an accurate effect of random calling on exam scores.

When pooling the data across all units of analysis, random call had a positive correlation with exam performance, which may be explained by a higher incidence of random call (mean = 1.49) and a larger variation (SD = 3.16). The more frequent use of random call across all courses may have contributed to students' comfort level with this PORTAAL practice which may help to reduce their anxiety. This could have allowed students to use more of their working memory to learn course material rather than dealing with feelings of anxiety.

**Explaining alternative answers.**   Providing opportunities for students to explain alternative answers to in-class questions was correlated with increases in exam scores across the three SEM analyses. We defined and used the code 'alternative answers' each time the instructor or student explained why incorrect or partially incorrect answers were incorrect, or if they provided multiple correct answers to a problem. Explaining alternative answers provides students with the opportunity to compare and contrast other possible answers to a question. This often

happens when explaining why the distractors in a multiple-choice option are incorrect or when discussing an open-ended question that has multiple correct answers. This practice highlights for students key distinguishing features of concepts [65] and is categorized as a type of Constructive activity in the ICAP framework [26]. Therefore, identifying incorrect answers during class may provide students with valuable test-taking skills and foster deeper conceptual understanding that can be used effectively on exams.

**Opportunity for students to work alone.** Allowing students to work alone prior to answering a question had opposite associations with exam scores depending on the Bloom's level of exams, positive for high Bloom's exams and negative for exams at low Bloom's levels. High Bloom's questions are cognitively demanding as they ask students to analyze, synthesize, or evaluate course topics which requires focused attention. In a study on how the activity of working alone on an in-class question impacted their experiences, students reported that they felt it was very important to form their own opinion without the influence of others prior to answering a question [66]. Students also indicate that having the opportunity to collect and formulate their thoughts prior to peer interaction allowed them to have a more fruitful discussion as they had more to contribute [66]. These results imply that working alone allows students the time to generate and formulate their thinking on the question and therefore is a type of constructive engagement with the material. Furthermore, reinforcing this type of cognitive behavior during class time could carry over to exams and encourage students to collect and organize their thoughts prior to answering.

Courses that test at lower Bloom's levels of recall, comprehension, and rote application often use more multiple-choice, matching, or fill-in the blank type questions [35] where students are unable to reason their way to the correct answer if they have not already memorized the required information. Research [67] indicates that students adapt their study strategies to meet but not exceed the challenge level of the exam. In other words, exams drive student learning. It is possible that in courses that test at lower Bloom's levels, students may not benefit from reasoning through the problem on their own as the answer is usually a factual piece of information that they either know or they do not know. In this case, taking time to work alone may cause more frustration if students are unable to answer the question on their own. We acknowledge that working alone should benefit students at all Bloom's levels of exams, and further research should investigate if there are components of this practice that could be associated with decreased performance on low Bloom's exams.

**Total student time in class.** Across all units of analysis, the total amount of time students were actively engaged in course material was positively correlated with exam performance when controlling for Bloom's level. Student time was recorded as any time during the class in which the instructor is not actively lecturing, giving instructions, or explaining the answer to a question. Total student time includes time that students are working individually, with a group of peers, or debriefing the answer to a question to the whole class.

'Student time' is the broadest of the codes that PORTAAL defines. It can be interpreted as a general measure of how student-centered versus instructor-centered the classroom is. Based on the findings of numerous studies showing the positive value of transforming STEM courses from traditional passive lecture to higher student engagement, The National Science Foundation's Vision and Change report [11] created a list of action items for faculty to undertake. Four of the eight action items included the following: engage students as active participants and not passive recipients in all undergraduate biology courses, use multiple modes of instruction in addition to traditional lecture, facilitate student learning within a cooperative context, and give students ongoing, frequent, and multiple forms of feedback on their progress. Our results provide empirical support for these suggested action items as we found that it may be beneficial for instructors to increase the amount of time students are actively working on

course material and decrease the amount of time they are delivering content. This does not mean that instructors give up their role in the classroom, but rather that they put their energy into designing class activities that provide students with the opportunity to create and deepen their understanding. This increased student engagement with course material may be achieved by incorporating more of the PORTAAL practices more often in their course. It may be encouraging for instructors to realize that implementing any of the PORTAAL practices they feel comfortable using will increase total student time in learning activities which has a positive correlation with exam performance.

**Performance on cognitively challenging exams.**   In classes with exams at higher Bloom's levels, two other PORTAAL practices correlated with improved exam scores: students explaining the logic underpinning answers and receiving positive feedback from the instructor. Students 'explaining their answers' were coded when a student or group of students provided reasoning behind their answer in front of the whole class. 'Positive feedback' from the instructor was coded any time the instructor used affirming language to praise the work of an individual student or the whole class.

Reasoning ability is crucial to answering cognitively challenging exam questions, and the more opportunities a student gets to practice this skill under the guidance of the instructor, the more likely the student will succeed on the exam. By having students explain the logic underlying their answers to the whole class while debriefing the question, instructors are reaffirming the necessity of generating and articulating a more sophisticated understanding of the material. In related findings, Knight et al. [68] found that when the instructor in an upper-division Biology course prompted students to explain their reasoning, more reasoning was noted in transcripts of peer discussions and during report outs. These student groups also more often arrived at the correct answer.

Providing positive feedback to students also creates a supportive climate and is observed more often in high-achieving classes [69]. Similarly, teacher confirmation of student work has been shown to increase student participation in class and greater self-reported use of study behaviors associated with cognitive learning [70]. Collectively, the additional practice of using logic to answer in-class questions and the supportive climate created by positive feedback may contribute to improved performance on cognitively challenging exams.

**Practical implementation of the PORTAAL practices.**   A convenient way for instructors to incorporate the PORTAAL practices the model predicted to significantly increase exam scores is by using in-class questions that students answer using a personal response system (i.e., clickers or Poll Everywhere) [71–73]. After posing the question, the instructor allows students a short period to work **alone** on the question and then enter their answer on their voting device. Without showing the voting results, the instructor asks the students to work in a **small group** with nearby peers to discuss and defend their answer. After a period of time, students re-enter their final answer. When debriefing the answer, the instructor **randomly calls** on a student or group of students to provide their answer and the reasoning underlying their answer. The instructor can also call on students to explain **alternative answers**. Activities like these can increase the amount of **student time** during class by allowing students to explore the problem on their own, with their peers, and to finally share their answer and reasoning with the whole class. Following this strategy could help faculty feel more comfortable and confident implementing PORTAAL practices.

## Limitations

Our retrospective study has an observational design, as faculty decided which teaching practices to use, and students selected which courses to enroll in. Therefore, there is the potential

for student academic performance to be influenced by factors that we have not measured and/ or that are outside our variables of interest (e.g., scheduling issues with other required courses in their major, work hours, or their preference for a certain course or instructor). We also recognize that the results from this study are based on a sample of biology instructors and students from one large research university and may not be generalizable to all institutions, faculty, or students.

We only collected and coded three classroom videos for the majority of the units of analysis. A study published after the first three academic years of data collection suggests that at least four observations are necessary for reliable characterization [41]. Future studies should include at least four classroom observations to more accurately measure teaching practices during an entire course.

Course exams can be of varying quality and often reflect the instructors' philosophies and beliefs about the role of assessment. There are possible third variable explanations for the correlations we found between PORTAAL practices and exam scores, including instructors' grading methods or how difficult they made the exams. We attempted to control for the cognitive challenge of exams by using weighted Bloom's level of exams as a covariate in our SEM. Though the processes of writing exams and grading are less than ideal, exam performance is often the main factor that determines if a student proceeds to the next level of their academic career. We also realize that for many courses, other formative assessments or laboratory work may contribute to a student's grade. However, the design and grading of these other forms of assessment was too varied for us to use as a proxy for learning.

To minimize the confounding effects in our study, we included the cognitive challenge of exams as measured by Bloom's level, students' GPA at the start of the term, and students' demographic information in our models. We did not control for Bloom's level of individual exam questions as we were not able to collect data by exam question by student. However, we were able to detect functional differences using the weighted Bloom's level of the entire exam. In future studies, we recommend strengthening the measures of exam and student characteristics by calculating item difficulty and student ability. These can only be calculated by exam question level analysis (e.g., using Rasch modeling), which would necessitate instructors to collect performance on each exam question for each student.

Our findings do not preclude the possibility that other teaching practices may have significant relationships with academic performance. Practices were included in the original PORTAAL rubric based on evidence that when only that practice was added to the current teaching method used in the classroom, that practice improved student academic performance. Because we included multiple practices in the same model, this could reduce our ability to detect significant effects of each of the PORTAAL practices. Given our sample size, we are unable to determine if implementation of multiple PORTAAL practices work additively or synergistically via interaction effects.

## Conclusions

Considerable evidence supports the claim that implementing active learning in undergraduate STEM classrooms improves student academic performance [1–6]. However, active learning encompasses a wide variety of teaching practices and determining which type, intensity or duration of active learning is effective in improving student performance has been elusive. Using PORTAAL to document classroom teaching practices allowed us to have a fine-grained analysis that–in conjunction with categorizing Bloom's level of exam questions–allowed us to isolate the practices that are correlated with performance on all exams as well as performance on exams at high Bloom's levels. Our study is the first to demonstrate a correlation between

the intensity or duration of specific evidence-based PORTAAL practices and exam performance. Across all units of analysis when controlling for Bloom's level of exams and students' GPA, implementation of small group activities, randomly calling on students for answers, offering alternative answers to questions, and total amount of time in class when students were actively engaged with course material were positively associated with exam scores. We also found that small group work offered a small additional benefit to women in courses with lower Bloom's level exams. These four practices were also predicted to increase scores on exams at lower Bloom's levels. If the faculty's goal is to increase student performance on cognitively demanding exams, they should consider increasing the number of times students work alone, explain their answers, and remind themselves to provide positive feedback to their students.

## Supporting information

**S1 Table. Instructor information.**
(PDF)

**S2 Table. Number of class videos coded using PORTAAL for 46 units of analysis.**
(PDF)

**S3 Table. Number of class videos coded using PORTAAL for final 42 units of analysis used in SEM.**
(PDF)

**S4 Table. LPA fit statistics.** BIC = Bayesian Information Criterion; ABIC = Adjusted BIC; BLRT = Bootstrap Likelihood Ratio Test; LMRT = Lo-Mendell-Rubin Adjusted Likelihood Ratio Test.; BF = Bayes Factor; cmP = Correct Model Probability; SIC = Schwarz Information Criterion. Best fit statistics are in boldface.
(PDF)

**S5 Table. Correlation matrix of 14 PORTAAL practices.** $^*p < 0.05$, $^{**}p < 0.01$, $^{***}p < 0.001$.
(PDF)

**S6 Table. Average intensity (instances) or duration (minutes) of 21 evidence-based teaching practices for all units of analysis.** Mean values with standard deviation in parentheses. $n$ all units = 46, $n$ high Bloom's = 24, $n$ low Bloom's = 22.
(PDF)

**S1 Fig. PORTAAL rubric.**
(PDF)

**S2 Fig. Histogram of Bloom's level of biology exams.** Distribution of the weighted Bloom's scores of biology exams for all 46 units of analysis. BL = Weighted Bloom's level of exams. Median = 53.8, SD = 8.00. Median value is shown as a blue vertical line.
(TIF)

**S3 Fig. SEM path diagram of full model tested for research question 1 with all PORTAAL practices and demographic interactions.** All 14 PORTAAL practices with demographic variables as mediators were tested but are only shown for small group activities to reduce complexity of the figure. PORTAAL practice boxes are in red. Covariates, interactions between demographic variables, and mediator boxes are in blue.
(TIF)

**S1 File. Daily PORTAAL values for all units of analysis.**
(XLSX)

**S2 File. Average student exam scores, demographic information, and mean PORTAAL values for all student data points.** Includes mean values based on the daily observations of the 14 PORTAAL practices included in SEM analyses for each unit of analysis. Binary gender coded as 1 = male, 0 = female. EOP and URM coded as 1 = EOP/URM, 0 = non-EOP/non-URM.
(XLSX)

## Acknowledgments

We thank Sarah Eddy and Mercedes Converse for their previous research on PORTAAL which this study builds on. We also thank the undergraduate researchers (E. Swanberg, P.J. Kambhiranond, A. Immel, C. Meng, and J. McAleer) who coded the class recordings. We also appreciate feedback on previous drafts from the Biology Education Research Group.

## Author Contributions

**Conceptualization:** Jennifer H. Doherty, Mary Pat Wenderoth.

**Data curation:** Sungmin Moon, Mallory A. Jackson.

**Formal analysis:** Sungmin Moon.

**Funding acquisition:** Jennifer H. Doherty, Mary Pat Wenderoth.

**Investigation:** Sungmin Moon, Mallory A. Jackson.

**Methodology:** Sungmin Moon.

**Project administration:** Jennifer H. Doherty, Mary Pat Wenderoth.

**Supervision:** Jennifer H. Doherty, Mary Pat Wenderoth.

**Visualization:** Sungmin Moon, Mallory A. Jackson.

**Writing – original draft:** Sungmin Moon, Mallory A. Jackson, Mary Pat Wenderoth.

**Writing – review & editing:** Sungmin Moon, Mallory A. Jackson, Jennifer H. Doherty, Mary Pat Wenderoth.

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
