## [Decision Letter · Decision Letter 0]

26 Aug 2021

PONE-D-21-24236

Evidence-based teaching practices correlate with increased exam performance in biology

PLOS ONE

Dear Dr. Wenderoth,

Thank you for submitting your manuscript to PLOS ONE. After careful consideration, we feel that it has merit but does not fully meet PLOS ONE’s publication criteria as it currently stands. Therefore, we invite you to submit a revised version of the manuscript that addresses the points raised during the review process.

As you will see from their comments below, the two expert reviewers are generally enthusiastic about the manuscript and the research and, from my own reading of the manuscript, I concur. However, both reviewers raise concerns that should be addressed before a final decision can be made. Of particular note, Reviewer 1 raises the concern that there is an implicit assumption that the exams, their grading and the instructors beliefs and teaching practices are not correlated to begin with. The Reviewer suggests ways this concern could be addressed and I believe these should be doable without further data collection. Reviewer 2, Dr. Jamie Jensen, raises the concerns that (a) Bloom's levels and difficulty are being treated as equivalent when that is not always the case and that (b) the explanation of why some predictors should be negative at some levels of Bloom's taxonomy and simultaneously positive at other levels is lacking. In your revision, you should address these concerns as well as all the other comments provided by both reviewers.

Finally, in your Data Availability statement you state that all data is available without restriction. However, I did not see any link to a data repository and the data provided in the Supplementary Materials seems to be only group averaged results or statistical tables. To comply with PLOS ONE Data Availability policy, I encourage you to make the raw coding data available through a repository in such a way that other researchers are able to replicate your findings and extend them if desired.

Should you resubmit a revision, I will send the new version to some or all of the same reviewers.

We look forward to receiving your revised manuscript.

Kind regards,

Paulo F. Carvalho

Academic Editor

PLOS ONE

Journal Requirements:

Reviewers' comments:

Reviewer's Responses to Questions

**Comments to the Author**

1. Is the manuscript technically sound, and do the data support the conclusions?

Reviewer #1: Partly

Reviewer #2: Yes

2. Has the statistical analysis been performed appropriately and rigorously? 

Reviewer #1: I Don't Know

Reviewer #2: Yes

3. Have the authors made all data underlying the findings in their manuscript fully available?

Reviewer #1: Yes

Reviewer #2: Yes

4. Is the manuscript presented in an intelligible fashion and written in standard English?

Reviewer #1: Yes

Reviewer #2: Yes

5. Review Comments to the Author

Reviewer #1: I would really love for the analysis and conclusions of this paper to be correct, but as it stands, there is one very fundamental potential flaw in it that must be addressed. It is possible that there is data showing that this potential flaw is not real, but if that is the case, it needs to be in the paper. However, if that is not the case, I believe that the work is fundamentally flawed and, unfortunately, not appropriate for publication. However, if this problem can be addressed by a small amount of additional data and explanation, then I could provide a more detailed and largely favorable review. However, I need to see what they say about this issue before digging deeply into the rest of the analysis for further review, since it is so relevant. So I would hope they might revise and resubmit.

The potential flaw is the implicit assumption of no correlation between the nature of the exams and their grading on one hand, and the views and teaching practices of the instructor on the other. They do structural equation modeling looking at how exam scores across a variety of courses correlate with various teaching practices, with the assumption that these practices result in better exam performance. However, if the exams and grading policies are set by the individual instructors, then an alternative and likely more plausible explanation of their results is simply that teachers who are more inclined to use the various PORTALL practices are also likely to give exams on which students get better grades. In that case, all they are showing is an indirect and not very interesting correlation, teacher attitudes about learning impact both their teaching and exam practices. It says nothing about the impact of teaching practices on learning.

My concern is not an abstract speculation. I have done a moderate amount of analysis of university science exams and their grading, and I have seen how wildly variable they are in character and quality. Right now, I am trying to get two courses changed in which the instructors are adamant about neither letting students see the correct answers to the exam questions and/or the basis on which their individual exams were graded. I know of other courses where the exams are largely puzzles covering material related to but not covered in class, and others where the grading makes no logical sense. In these, and other examples I could unfortunately give, the exam score is largely unrelated to mastery of the material, and instead primarily depends on figuring out the idiosyncrasies of the instructor. I have also seen that generally those faculty who are inclined to use more active learning practices are also more inclined to have more meaningful and transparent exams and grading.

In this paper, the only description of the exams is to say they are given by the different instructors, and the researchers rated them according to Bloom’s level. It is true that higher Bloom’s level questions are generally associated with greater difficulty and lower scores, that is often only a small part of what determines the score on an exam question. The many other instructor idiosyncrasies matter a great deal, including such simple features as whether the instructor believes the average grade should be 50% or 80%, again, a difference I have observed in practice.

What I have discussed are examples of the extreme but actual cases where exam scores are strongly dependent on instructor attitudes, and hence tend to also correlate with their attitudes about teaching practices. That is not always true, however. I have also seen departments where the exams and grading policies were tightly controlled by the department. Instructors were quite constrained as to what questions they gave and how they were graded, in some cases even having exams of the large courses created by a committee separate from the course instructors. If something like that was the case for the courses analyzed in this work, my concerns would vanish.

So I cannot claim that this work is flawed, only that this question of the nature of the exams and grading and how sensitive and variable these are according to individual instructor preferences needs to be addressed carefully. As I said above, I would be happier if they had evidence showing the exams were largely independent of instructor idiosyncrasies, and so the claims of the paper were justified, but until this is shown, the paper is not suitable for publication.

Reviewer #2: This article looks at the relationship between PORTAAL practices (evidence-based pedagogies) and student exam performance taking into account Bloom’s level of exams and several demographic factors of students. It shows a relationship between various PORTAAL practices and performance and some that even differential impact men and women. It is an exciting study that has potential to shape the way in which we, as teachers, design our active learning classrooms. I have a few comments that I think will help to strengthen this paper. I am listing them below in no particular order of importance (just in the order that I encountered them while reading).

1. On page 5, line 127, the authors make the claim that “Bloom level of exams can then be used to control for the cognitive challenge leve of exams, as students would be expected to perform better on easier (low Bloom) versus harder (high Bloom) exam questions. I worry a little bit about the way this is stated and it appears the authors are conflating Blooms level with difficulty. I would agree that the lower levels of Blooms have been considered less cognitively challenging and higher levels of Blooms more cognitively challenging (often referred to as LOCS and HOCS; see Crowe, A., Dirks, C., Wenderoth, M. P. (2008). Biology in Bloom: Implementing Bloom's Taxonomy to Enhance Student Learning in Biology. [J] Cbe-Life Sciences Education, 7, 4: 368-381. doi:10.1187/cbe.08-05-0024 and Zoller, U. (1993). Are lecture ad learning compatible? Maybe for LOCS: Unlikely for HOCS [J]. Journal of Chemical Education, 70, 3: 195-197. doi: 10.1021/ed070p195). However, cognitive difficult is often mistakenly conflated with performance (see Lemons, P. P., Lemons, J. D. (2013). Questions for Assessing Higher-Order Cognitive Skills: It's Not Just Bloom's. [J] Cbe-Life Sciences Education, 12, 1: 47-58. doi:10.1187/cbe.12-03-0024 and Wyse, A. E., Viger, S. G. (2011). How item writers understand depth of knowledge. [J] Educational Assessment, 16, 4: 185–206. doi: 10.1080/10627197.2011.634286).

Often, performance doesn't reflect the actual cognitive difficulty assigned by Bloom's Taxonomy (see Momsen, J. L., Long, T. M., Wyse, S. A., Ebert-May, D. (2010). Just the Facts? Introductory Undergraduate Biology Courses Focus on Low-Level Cognitive Skills. [J] Cbe-Life Sciences Education, 9, 4: 435-440. doi:10.1187/cbe.10-01-0001), as many other factors may play a role in that difficulty (see Jensen, JL, Phillips, AJ, & Briggs, JC. (2019). Beyond Bloom’s: Students’ Perception of Bloom’s Taxonomy and its Convolution with Cognitive Load. Journal of Psychological Research, 1(1): 1-9.). I might just rephrase this to say that the Blooms level can be a proxy for cognitive challenge but may or may not reflect performance.

On the other hand, an easy way to test this is to do a quick analysis between assigned bloom level (low v high) and performance on a few selected tests to see if you can indeed see a strong and robust relationship between the two. You would probably need to do this individually for each instructor as exam writing styles and class structures likely play a role in whether or not cognitive level correlates with performance.

2. This is a very minor comment but I noticed it a couple of times in the paper. You are misusing colons. Colons only follow a complete sentence. So, you can say, "...limited to the following:", but not “…limited to:”. Alternatively, you can leave the colon out. Colons never come in the middle of a sentence or thought.

3. On page 10, line 239, under Bloom categorization of exam questions, can you clarify how many? and did all researcher rate all problems or did you need to establish IRR and then everyone divided and conquered?

4. I found Table 3 to be just a little confusing being exposed to it prior to your results from the low and high analysis. Perhaps you could add a column heading that said, “First Analysis – all together” and then “Second analysis – divided by Blooms level” or something like that. I just had so many questions until I realized this table applied also to result further down.

5. Page 16, line 344, you say, “…divide the standard deviation of the exam scores…” Instead, say, "we divided". As it reads now, it sounds more like a list of instructions for the reader.

6. I would strongly encourage you to comment in the RESULTS section on the negative coefficients. It was very confusing and I didn’t begin to make sense of it until the Discussion. Please mention it in the results.

7. On that same note, however, I am not convinced by the explanations in the discussions for why these should be negatively correlated in some instances and positively correlated in others (for some of these PORTAAL practices). Let me give some examples:

a. For example, with random call – it helped in high level but hurt in low level. Your explanation for why it helped in high level seems adequate. But, the reasoning for why it hurt in low level seems to be a reason that would apply to both high and low (anxiety). Why would this differentially impact low-level? Is there some kind of evidence that would suggest that instructors who used low-level exams invoked more anxiety? I’m not sure this is an adequate explanation.

b. The same is true of working alone. I can see how (and I totally buy your explanation) working alone first on high-level items would be a benefit. But, I don’t see how working alone first would actually harm someone on a low-level question. You sort of hinted at maybe taking time to work along left less time to answer other questions in class, but I’m not sure that’s very convincing. If anything, I would predict no effect. Why would it hurt them?

8. Page 22, line 477, sentence starting with “We defined and used the code alternative answers each time…” I had to read it like 7 times before I understood what you were saying! It would help if you put the code names in quotes or used italics or something. It took me forever to figure out that ‘alternative answers’ was a code name.

9. Throughout the paper, I'm struggling a little to reconcile the two-level distinction you use in the intro and discussion (high blooms and low blooms) with the three-level distinction you used to code items (low--remember/understand, medium--apply/analyze, and high--evaluate/synthesis). Which was used in the analyses? Can you be a little clearer with this?

10. Page 23, line 50, you make a comment about eastern and western thought and East Asian cultures. This is a weird addition to the paper considering you did not look at East Asian cultures vs. Westerners. It just seems to come out of the blue. I would just leave it out.

This is a well-written paper and a fascinating study. I look forward to seeing the edited draft!

6. PLOS authors have the option to publish the peer review history of their article (what does this mean?). If published, this will include your full peer review and any attached files.

Reviewer #1: No

Reviewer #2: **Yes: **Jamie L. Jensen

---

## [Author Response · Author response to Decision Letter 0]

15 Sep 2021

RESPONSES

Finally, in your Data Availability statement you state that all data is available without restriction. However, I did not see any link to a data repository and the data provided in the Supplementary Materials seems to be only group averaged results or statistical tables. To comply with PLOS ONE Data Availability policy, I encourage you to make the raw coding data available through a repository in such a way that other researchers are able to replicate your findings and extend them if desired.

We are unable to provide the raw coding data as this data is on paper in notebooks and contains information that identifies instructors. We have supplied in the supporting information the aggregated mean values for each of the PORTAAL practices for each of the daily observations that were recorded on paper. We will also provide the full-data set used for our SEM analysis which includes all 14,000+ student data points for the 46 units of analysis. 

We note that the grant information you provided in the ‘Funding Information’ and ‘Financial Disclosure’ sections do not match.

We will correct this oversight when we resubmit.

Reviewer #1: I would really love for the analysis and conclusions of this paper to be correct, but as it stands, there is one very fundamental potential flaw in it that must be addressed. It is possible that there is data showing that this potential flaw is not real, but if that is the case, it needs to be in the paper. However, if that is not the case, I believe that the work is fundamentally flawed and, unfortunately, not appropriate for publication. However, if this problem can be addressed by a small amount of additional data and explanation, then I could provide a more detailed and largely favorable review. However, I need to see what they say about this issue before digging deeply into the rest of the analysis for further review, since it is so relevant. So I would hope they might revise and resubmit.

The potential flaw is the implicit assumption of no correlation between the nature of the exams and their grading on one hand, and the views and teaching practices of the instructor on the other. They do structural equation modeling looking at how exam scores across a variety of courses correlate with various teaching practices, with the assumption that these practices result in better exam performance. However, if the exams and grading policies are set by the individual instructors, then an alternative and likely more plausible explanation of their results is simply that teachers who are more inclined to use the various PORTALL practices are also likely to give exams on which students get better grades. In that case, all they are showing is an indirect and not very interesting correlation, teacher attitudes about learning impact both their teaching and exam practices. It says nothing about the impact of teaching practices on learning

My concern is not an abstract speculation. I have done a moderate amount of analysis of university science exams and their grading, and I have seen how wildly variable they are in character and quality. Right now, I am trying to get two courses changed in which the instructors are adamant about neither letting students see the correct answers to the exam questions and/or the basis on which their individual exams were graded. I know of other courses where the exams are largely puzzles covering material related to but not covered in class, and others where the grading makes no logical sense. In these, and other examples I could unfortunately give, the exam score is largely unrelated to mastery of the material, and instead primarily depends on figuring out the idiosyncrasies of the instructor. I have also seen that generally those faculty who are inclined to use more active learning practices are also more inclined to have more meaningful and transparent exams and grading.

In this paper, the only description of the exams is to say they are given by the different instructors, and the researchers rated them according to Bloom’s level. It is true that higher Bloom’s level questions are generally associated with greater difficulty and lower scores, that is often only a small part of what determines the score on an exam question. The many other instructor idiosyncrasies matter a great deal, including such simple features as whether the instructor believes the average grade should be 50% or 80%, again, a difference I have observed in practice.

What I have discussed are examples of the extreme but actual cases where exam scores are strongly dependent on instructor attitudes, and hence tend to also correlate with their attitudes about teaching practices. That is not always true, however. I have also seen departments where the exams and grading policies were tightly controlled by the department. Instructors were quite constrained as to what questions they gave and how they were graded, in some cases even having exams of the large courses created by a committee separate from the course instructors. If something like that was the case for the courses analyzed in this work, my concerns would vanish.

So I cannot claim that this work is flawed, only that this question of the nature of the exams and grading and how sensitive and variable these are according to individual instructor preferences needs to be addressed carefully. As I said above, I would be happier if they had evidence showing the exams were largely independent of instructor idiosyncrasies, and so the claims of the paper were justified, but until this is shown, the paper is not suitable for publication.

Thank you for helping us better clarify the exam data for our audience. We agree with reviewer #1 that exams are a less than perfect means to assess the learning of students and have many drawbacks which reviewer #1 has identified. To address this issue we have added the following thoughts to the METHODS and LIMITATIONS sections of the paper.

This research is a retrospective study of the teaching methods used by faculty at an R1 in the Northwest. At this university all these classes were videotaped using lecture capture technology installed in the classrooms and these recordings are archived. Faculty involved in the study were not involved in any formal professional development and were not specifically trained in the use of the teaching practices associated with PORTAAL. Some of the faculty had read the PORTAAL paper as part of a faculty learning community where multiple DBER articles were discussed over a two year period. Most of the class videos were of classes taught prior to the publication of the PORTAAL paper. Faculty were merely asked if we could review videos of past classes to determine if a new tool (PORTAAL) we were developing could differentiate between courses using various levels of active learning. As this is a retrospective study, compositions of exams in these courses could not be changed to alter student academic performance.

Ours is a retrospective study as we were able to use archived recordings of class sessions in our analysis. Therefore the faculty wouldn’t have been able to change exam questions or exam performance as all this material was also archived. We agree that exams (summative assessments) can be of varying quality and certainly do reflect the idiosyncrasies of the instructor. Though the process of exam writing and grading are often less ideal than we would desire, exam performance is often the main factor that determines if a student proceeds to the next level of their academic career. We also realize that for many courses, other formative assessments or laboratory work may contribute to a student’s grade and we determined that the design and grading of these other assessment forms was too varied for us to use as a proxy for learning.

Reviewer #2: This article looks at the relationship between PORTAAL practices (evidence-based pedagogies) and student exam performance taking into account Bloom’s level of exams and several demographic factors of students. It shows a relationship between various PORTAAL practices and performance and some that even differential impact men and women. It is an exciting study that has potential to shape the way in which we, as teachers, design our active learning classrooms. I have a few comments that I think will help to strengthen this paper. I am listing them below in no particular order of importance (just in the order that I encountered them while reading).

1. On page 5, line 127, the authors make the claim that “Bloom level of exams can then be used to control for the cognitive challenge level of exams, as students would be expected to perform better on easier (low Bloom) versus harder (high Bloom) exam questions. I worry a little bit about the way this is stated and it appears the authors are conflating Blooms level with difficulty. I would agree that the lower levels of Blooms have been considered less cognitively challenging and higher levels of Blooms more cognitively challenging (often referred to as LOCS and HOCS; see Crowe, A., Dirks, C., Wenderoth, M. P. (2008). Biology in Bloom: Implementing Bloom's Taxonomy to Enhance Student Learning in Biology. [J] Cbe-Life Sciences Education, 7, 4: 368-381. doi:10.1187/cbe.08-05-0024 and Zoller, U. (1993). Are lecture and learning compatible? Maybe for LOCS: Unlikely for HOCS [J]. Journal of Chemical Education, 70, 3: 195-197. doi: 10.1021/ed070p195). However, cognitive difficult is often mistakenly conflated with performance (see Lemons, P. P., Lemons, J. D. (2013). Questions for Assessing Higher-Order Cognitive Skills: It's Not Just Bloom's. [J] Cbe-Life Sciences Education, 12, 1: 47-58. doi:10.1187/cbe.12-03-0024 and Wyse, A. E., Viger, S. G. (2011). How item writers understand depth of knowledge. [J] Educational Assessment, 16, 4: 185–206. doi: 10.1080/10627197.2011.634286).

Often, performance doesn't reflect the actual cognitive difficulty assigned by Bloom's Taxonomy (see Momsen, J. L., Long, T. M., Wyse, S. A., Ebert-May, D. (2010). Just the Facts? Introductory Undergraduate Biology Courses Focus on Low-Level Cognitive Skills. [J] Cbe-Life Sciences Education, 9, 4: 435-440. doi:10.1187/cbe.10-01-0001), as many other factors may play a role in that difficulty (see Jensen, JL, Phillips, AJ, & Briggs, JC. (2019). Beyond Bloom’s: Students’ Perception of Bloom’s Taxonomy and its Convolution with Cognitive Load. Journal of Psychological Research, 1(1): 1-9.). I might just rephrase this to say that the Blooms level can be a proxy for cognitive challenge but may or may not reflect performance.

On the other hand, an easy way to test this is to do a quick analysis between assigned bloom level (low v high) and performance on a few selected tests to see if you can indeed see a strong and robust relationship between the two. You would probably need to do this individually for each instructor as exam writing styles and class structures likely play a role in whether or not cognitive level correlates with performance.

Thank you for these comments and for taking the time to suggest all the valuable citations on this topic. We have incorporated the majority of these citations in the appropriate locations in the text.

1. Our current SEM results show that student exam performance is negatively correlated with Bloom’s level, which is in agreement with our earlier work (Freeman, Haak, & Wenderoth, 2011, see Fig. 2).

2. We have added the following thoughts to the paper.

To attempt to control for the varying cognitive challenge of exams across all courses, we categorized each exam question according to Bloom’s Taxonomy of Cognitive Domains (Bloom, 1956; Crowe et al., 2008). We realize that all attempts at categorizing exam questions have limitations. The challenge level of the exam can be influenced by many factors including but not limited to the following: the length and format of the question, wording of the question stem and distractors (Kibble, 2017) cognitive load of the question (Phillips, Briggs, & Jensen, 2019) and the alignment of the test with the level of instruction delivered (Lemons & Lemons, 2013; Wyse & Viger, 2011; Wiggins & McTighe, 2005).

Based on our previous research findings (Crowe et al., 2008; Freeman et al., 2011) and others (Stanger-Hall, 2012) we find Bloom’s Taxonomy a reasonable proxy for academic challenge level which we realize can differ from performance level. We acknowledge that questions at the lower level of Bloom’s taxonomy can be quite difficult for students and lead to lower performance (Momsen et al., 2010) as these questions can address minutia or require knowledge of very specific and often obscure differences between answer choices. However, previous research indicates a negative correlation between Bloom’s level and student exam performance (Freeman, Haak, & Wenderoth, 2011, see Fig. 2). Given these findings, it is important to determine how teaching methods impact student exam performance on exams at both low and high Bloom’s levels. As most faculty indicate on their syllabi that they want students working at higher cognitive levels (Momsen et al., 2010) in their courses, it is important to determine which teaching practices provide the best type of practice needed for students to excel at these higher levels of cognition.

In any case, it is important to determine how teaching methods impact student exam performance on exams at both HOC and LOC levels whether these exams are easy or not.

2. This is a very minor comment but I noticed it a couple of times in the paper. You are misusing colons. Colons only follow a complete sentence. So, you can say, "...limited to the following:", but not “…limited to:”. Alternatively, you can leave the colon out. Colons never come in the middle of a sentence or thought.

Thank you for this comment. We have searched the document for colons and either removed them or corrected the grammatical use of the colon.

3. On page 10, line 239, under Bloom categorization of exam questions, can you clarify how many? and did all researcher rate all problems or did you need to establish IRR and then everyone divided and conquered?

Thank you for this comment. We have added more description of the coding process to the methods section. 

“Two trained researchers (authors M.P.W. and M.A.J.) with expertise in biology independently coded all exam questions using the six levels of Bloom’s Taxonomy from a Bloom level 1 (knowledge) to Bloom level 6 (evaluation). The researchers then met to reconcile any differences in their codes and come to one-hundred percent consensus.”

4. I found Table 3 to be just a little confusing being exposed to it prior to your results from the low and high analysis. Perhaps you could add a column heading that said, “First Analysis – all together” and then “Second analysis – divided by Blooms level” or something like that. I just had so many questions until I realized this table applied also to result further down.

We have made the suggested changes to Table 3.

5. Page 16, line 344, you say, “…divide the standard deviation of the exam scores…” Instead, say, "we divided". As it reads now, it sounds more like a list of instructions for the reader.

We have made this change to the text.

6. I would strongly encourage you to comment in the RESULTS section on the negative coefficients. It was very confusing and I didn’t begin to make sense of it until the Discussion. Please mention it in the results.

Thank you for helping us clarify this point. We have added the following text in the RESULTS section under research question 2.

“In units of analysis with lower Bloom’s level exams on average, three PORTAAL practices were significantly positively correlated with student exam scores: small group activities, explaining alternative answers, and total student time. Two PORTAAL practices showed a significant negative correlation with exam scores: random call and students working alone.”

7. On that same note, however, I am not convinced by the explanations in the discussions for why these should be negatively correlated in some instances and positively correlated in others (for some of these PORTAAL practices). Let me give some examples:

a. For example, with random call – it helped in high level but hurt in low level. Your explanation for why it helped in high level seems adequate. But, the reasoning for why it hurt in low level seems to be a reason that would apply to both high and low (anxiety). Why would this differentially impact low-level? Is there some kind of evidence that would suggest that instructors who used low-level exams invoked more anxiety? I’m not sure this is an adequate explanation.

We have expanded our explanation of the seemingly contradictory findings for random call to include the following text. 

“However, using random call was significantly negatively correlated with student performance when exams were at lower Bloom’s levels. Research has shown that random call can induce some anxiety in students, which may decrease students’ sense of safety in the classroom and negatively impact exam performance [59,60]. There are multiple ways to implement random call that may alleviate the anxiety [60] and instructors need to determine the procedure that best matches the characteristics of their student population. As instructors in these courses had a low incidence of random call (mean = 0.38 times per class, SD = 1.55), it is possible that students did not have enough opportunities to become comfortable sharing their answers with the whole class, and instead felt anxious about random call. However, this result could also be due to not enough variation in our sample of courses with low Bloom’s exams to detect an accurate effect of random calling on exam scores.

When pooling the data across all units of analysis, random call had a positive correlation with exam performance, which may be explained by a higher incidence of random call (mean = 1.49) and a larger variation (SD = 3.16). The more frequent use of random call across all courses may have contributed to students’ comfort level with this PORTAAL practice which may help to reduce their anxiety. This could have allowed students to use more of their working memory to learn course material rather than dealing with feelings of anxiety.”

b. The same is true of working alone. I can see how (and I totally buy your explanation) working alone first on high-level items would be a benefit. But, I don’t see how working alone first would actually harm someone on a low-level question. You sort of hinted at maybe taking time to work along left less time to answer other questions in class, but I’m not sure that’s very convincing. If anything, I would predict no effect. Why would it hurt them?

Thank you for pointing out these contradictory results. We have revised our discussion of working alone to include the following text.

“Research [67] indicates that students adapt their study strategies to meet but not exceed the challenge level of the exam. In other words, exams drive student learning. It is possible that in courses that test at lower Bloom’s levels, students may not benefit from reasoning through the problem on their own as the answer is usually a factual piece of information that they either know or they do not know. In this case, taking time to work alone may cause more frustration if students are unable to answer the question on their own. We acknowledge that working alone should benefit students at all Bloom’s levels of exams, and further research should investigate if there are components of this practice that could be associated with decreased performance on low Bloom’s exams.”

8. Page 22, line 477, sentence starting with “We defined and used the code alternative answers each time…” I had to read it like 7 times before I understood what you were saying! It would help if you put the code names in quotes or used italics or something. It took me forever to figure out that ‘alternative answers’ was a code name.

Thank you for this suggestion, We have added quotation marks to help delineate our coding terms.

9. Throughout the paper, I'm struggling a little to reconcile the two-level distinction you use in the intro and discussion (high blooms and low blooms) with the three-level distinction you used to code items (low--remember/understand, medium--apply/analyze, and high--evaluate/synthesis). Which was used in the analyses? Can you be a little clearer with this?

Thank you for this comment. We have included the following changes to the text in the methods section describing Bloom’s categorization of exam questions. 

“Given our sample size, we could not parse and analyze three Bloom’s levels of exams. Therefore, we grouped units into either low or high Bloom’s level exams in the structural equation models, using the median value of the units (median = 53.8) as the cutoff.”

10. Page 23, line 50, you make a comment about eastern and western thought and East Asian cultures. This is a weird addition to the paper considering you did not look at East Asian cultures vs. Westerners. It just seems to come out of the blue. I would just leave it out.

Thank you for this suggestion. We have removed this section from the text.

This is a well-written paper and a fascinating study. I look forward to seeing the edited draft!

Thank you for your very helpful comments and encouragement. We hope you are satisfied with the changes we have made to the text.

---

## [Decision Letter · Decision Letter 1]

12 Nov 2021

PONE-D-21-24236R1Evidence-based teaching practices correlate with increased exam performance in biologyPLOS ONE

Dear Dr. Wenderoth,

Thank you for submitting your manuscript to PLOS ONE. After careful consideration, we feel that it has merit but does not fully meet PLOS ONE’s publication criteria as it currently stands. Therefore, we invite you to submit a revised version of the manuscript that addresses the points raised during the review process. As you will see from the reviews appended below, Reviewer 1 points out that a critical issue is still present in the revised version of the manuscript: the findings might be due to what I'd call a third variable (e.g., that teachers who are more likely to use active learning practices are also more likely to be more lenient in their grading or create easier exams). Although, from my reading, I believe you were careful to not state causal claims and instead refer to the findings as correlations, I believe this issue should be addressed directly in the text. Thus, please address Reviewer 1's concern by directly stating in the text that there are possible 3rd variable explanations to the results (others are possible as well, such as more active learning being more likely in "easier" courses, or at lower level courses) and the steps you took (or could not take) to address this concern. Your previous edits do not directly address this because they focus on generic limitations of using existing data. Also, please make sure to change any lingering causal language to emphasize that the relations you found, although indicative, should not be interpreted as causal. To avoid an extended review process, I will aim to make a decision when I receive a revised version without sending the manuscript for further review.

We look forward to receiving your revised manuscript.

Kind regards,

Paulo F. Carvalho

Academic Editor

PLOS ONE

Journal Requirements:

Reviewers' comments:

Reviewer's Responses to Questions

**Comments to the Author**

1. If the authors have adequately addressed your comments raised in a previous round of review and you feel that this manuscript is now acceptable for publication, you may indicate that here to bypass the “Comments to the Author” section, enter your conflict of interest statement in the “Confidential to Editor” section, and submit your "Accept" recommendation.

Reviewer #1: (No Response)

Reviewer #2: All comments have been addressed

2. Is the manuscript technically sound, and do the data support the conclusions?

Reviewer #1: No

Reviewer #2: Yes

3. Has the statistical analysis been performed appropriately and rigorously? 

Reviewer #1: Yes

Reviewer #2: Yes

4. Have the authors made all data underlying the findings in their manuscript fully available?

Reviewer #1: Yes

Reviewer #2: Yes

5. Is the manuscript presented in an intelligible fashion and written in standard English?

Reviewer #1: Yes

Reviewer #2: Yes

6. Review Comments to the Author

Reviewer #1: The authors have not adequately addressed my previous concern. I do not think the evidence they present is adequate to support their conclusion. Their conclusion, which is what makes this paper notable, is that use of more active learning ("portaal practices") leads to improved exam performance. I hope that this is true, but it is at least as likely, if not more so, that what they have actually observed is simply that instructors that use more active learning grade easier.

Their conclusion rests on the assumption that the different instructors have some underlying equivalence in their testing and grading practices, which makes it meaningful to compare results across instructors and attribute differences in exam grades to differences in teaching practices. Given how arbitrary and idiosyncratic exam and grading practices are across faculty at US universities, in the absence of evidence that there is some level of consistency which makes this comparison across instructors meaningful, I do not see how their method and conclusions can be justified.

Reviewer #2: (No Response)

7. PLOS authors have the option to publish the peer review history of their article (what does this mean?). If published, this will include your full peer review and any attached files.

Reviewer #1: No

Reviewer #2: **Yes: **Jamie L. Jensen

---

## [Author Response · Author response to Decision Letter 1]

15 Nov 2021

RESPONSES 

Editor comment: As you will see from the reviews appended below, Reviewer 1 points out that a critical issue is still present in the revised version of the manuscript: the findings might be due to what I'd call a third variable (e.g., that teachers who are more likely to use active learning practices are also more likely to be more lenient in their grading or create easier exams). Although, from my reading, I believe you were careful to not state causal claims and instead refer to the findings as correlations, I believe this issue should be addressed directly in the text. Thus, please address Reviewer 1's concern by directly stating in the text that there are possible 3rd variable explanations to the results (others are possible as well, such as more active learning being more likely in "easier" courses, or at lower level courses) and the steps you took (or could not take) to address this concern. Your previous edits do not directly address this because they focus on generic limitations of using existing data. Also, please make sure to change any lingering causal language to emphasize that the relations you found, although indicative, should not be interpreted as causal.

Response: You will see from our track changes document that we have changed any lingering causal language. We have removed the word “impact” and replaced it with “correlated”. In the first paragraph of our SEM section of the Methods where we discuss variables that could be contributing to the residual variance we have added instructor exam grading methods. We have also directly added “possible 3rd variable explanations” in our limitation section. 

Our study is an attempt to do an in vivo study of how implementation of multiple PORTAAL practices in actual classroom settings across multiple large courses at an R1 institution would correlate with changes in student exam performance. As with all in vivo studies, we are constrained by the actual conditions that exist in the in vivo setting. We identified multiple factors that we controlled for in our analysis (incoming GPA of the student, gender, first generation status, minority status, academic challenge level of the exam, etc). What reviewer #1 is asking for, standardized grading across courses, is unreasonable to request in a truly in vivo study.

Reviewer #1 comment: The authors have not adequately addressed my previous concern. I do not think the evidence they present is adequate to support their conclusion. Their conclusion, which is what makes this paper notable, is that use of more active learning ("portaal practices") leads to improved exam performance. I hope that this is true, but it is at least as likely, if not more so, that what they have actually observed is simply that instructors that use more active learning grade easier.

Their conclusion rests on the assumption that the different instructors have some underlying equivalence in their testing and grading practices, which makes it meaningful to compare results across instructors and attribute differences in exam grades to differences in teaching practices. Given how arbitrary and idiosyncratic exam and grading practices are across faculty at US universities, in the absence of evidence that there is some level of consistency which makes this comparison across instructors meaningful, I do not see how their method and conclusions can be justified.

Response: We agree with reviewer #1 that exams and grading can be arbitrary and idiosyncratic across faculty at US universities yet exams are the very methods faculty have chosen to use as one means to assess student understanding of course material. In the department at the university in which this study was conducted, the overwhelming majority of the courses have lab associates that help to standardize exams and grading across quarters and across courses. This is the department’s attempt to provide consistency for the students as they move through the curriculum.

In our 2011 paper, we showed that as more active learning was incorporated into the course the academic challenge level of the exams actually increased, thus the exams got harder not easier. In this paper we attempted to control for the academic challenge level of the exam by categorizing each exam question based on Bloom’s taxonomy of Cognitive Domains and then controlled for Bloom level of exams. We have added to the limitation section of the paper Reviewer #1 concern about grading and we have removed all language that might hint at causality from the manuscript. Furthermore, in the manuscript, we are careful to state that our model predicted the results we are reporting.

---

## [Editor Report · Decision Letter 2]

17 Nov 2021

Evidence-based teaching practices correlate with increased exam performance in biology

PONE-D-21-24236R2

Dear Dr. Wenderoth,

We’re pleased to inform you that your manuscript has been judged scientifically suitable for publication and will be formally accepted for publication once it meets all outstanding technical requirements.

Kind regards,

Paulo F. Carvalho

Academic Editor

PLOS ONE
---

## [Editor Report · Acceptance letter]

19 Nov 2021

PONE-D-21-24236R2 

Evidence-based teaching practices correlate with increased exam performance in biology 

Dear Dr. Wenderoth:

I'm pleased to inform you that your manuscript has been deemed suitable for publication in PLOS ONE. Congratulations! Your manuscript is now with our production department. 

Kind regards, 

on behalf of

Dr. Paulo F. Carvalho 

Academic Editor

PLOS ONE